# Molecular Chaperones and Proteolytic Machineries Regulate Protein Homeostasis in Aging Cells

**DOI:** 10.3390/cells9051308

**Published:** 2020-05-24

**Authors:** Boris Margulis, Anna Tsimokha, Svetlana Zubova, Irina Guzhova

**Affiliations:** Institute of Cytology of the Russian Academy of Sciences, St. Petersburg 194064, Russia; margulis@incras.ru (B.M.); atsimokha@incras.ru (A.T.); egretta_julia@mail.ru (S.Z.)

**Keywords:** molecular chaperones, autophagy, ubiquitin-proteasomal system, aging

## Abstract

Throughout their life cycles, cells are subject to a variety of stresses that lead to a compromise between cell death and survival. Survival is partially provided by the cell proteostasis network, which consists of molecular chaperones, a ubiquitin-proteasome system of degradation and autophagy. The cooperation of these systems impacts the correct function of protein synthesis/modification/transport machinery starting from the adaption of nascent polypeptides to cellular overcrowding until the utilization of damaged or needless proteins. Eventually, aging cells, in parallel to the accumulation of flawed proteins, gradually lose their proteostasis mechanisms, and this loss leads to the degeneration of large cellular masses and to number of age-associated pathologies and ultimately death. In this review, we describe the function of proteostasis mechanisms with an emphasis on the possible associations between them.

## 1. Introduction

Aging is a process that, according to Maurice Chevalier, “isn’t so bad when you consider the alternative.” In numerous essays, aging or senescence is presented as a gradual dying process that involves a loss of cell integrity, leading to the dysfunction of cells or organs and eventually to their deaths. Importantly, aging as underlined by a majority of researchers is marked by an increased production of β-galactosidase, p21waf-p53 axis, γ-H2AX, advanced glycation end products, and other indicators that are also used as markers of senescence [1]. A few years ago, in their review, López-Otín et al. formulated nine hallmarks of aging, among which are genomic instability, deregulated nutrient sensing, mitochondrial dysfunction, cellular senescence, and loss of proteostasis [2]. The factors that accompany or induce senescence include, as the major ones, errors in mRNA translation, inappropriate protein modification (resulting from oxidative and reductive stresses), reduction in cell ionic homeostasis, inhibition of membrane function, and others [3].

There are different ways of analyzing changes occurring in aging, such as a comparison of the initial and final states of molecular/cellular structures related to protein homeostasis and estimation of those in dynamics. As such, the malfunction of cellular proteostasis machinery may imply the current improper state of protein synthesis/modification/transport/utilization machinery and/or inadequacy of its dynamics during the aging (e.g., instant response and capacity to improve those functions). Consequently, the absence of a well-arranged system to respond to proteotoxic pathogens, such as β-amyloid, α-synuclein, and mutant huntingtin, suggests the development of the corresponding age-related pathologies, namely Alzheimer’s disease, Parkinson’s disease, and Huntington’s disease, respectively [4].

Cell proteostasis machinery consists of several major parts aimed to monitor wrongly-assembled polypeptides and their complexes and includes molecular chaperones, autophagy, ubiquitin-proteasomal system of degradation (UPS), and unfolded protein response (UPR) [5]. Molecular chaperones are mostly represented by so-called heat shock proteins that are able to recognize misfolded polypeptides and correct their structure or transfer them to the UPS for degradation [4,6]. Autophagy encompasses at least four differently-arranged systems: macroautophagy, microautophagy, selective autophagy, and chaperone-mediated autophagy (CMA). The systems differ in their substrate targets, which include separate polypeptides or whole protein supramolecular structures such as mitochondria, and, correspondingly, in mechanisms of function [7,8]. The UPS targets unnecessary polypeptides, labels them with ubiquitin, and cleaves them in special particles called proteasomes [9]. The UPR is a stress response activated by the excess of unfolded proteins in the endoplasmic reticulum, and its uncontrolled activation is responsible for several pathologies, including cancer and metabolic, neurodegenerative, and inflammatory diseases [10] (Figure 1).

Recent studies show that the above proteostasis mechanisms lose their efficacy in aged cells or organisms. Particularly convincingly, this negative trend has been proven when analyzing *Caenorhabditis elegans*. The heat shock response, mitochondrial unfolded protein response, endoplasmic reticulum unfolded protein response, hypoxia response, SKN-1-mediated oxidative stress response, and the DAF-16-mediated stress response have all been shown to decline with age in nematodes, as demonstrated using the array of the corresponding reporter assays [4,11].

The compromise between cell death and survival is often linked to a state of proteostasis in a single cell and may be regulated by endogenous signaling. For instance, in tumor cells constantly experiencing the pressure of stressful factors, heat shock factor 1 (HSF1), a transcriptional activator, causes the enhancement of heat shock protein synthesis, which protects the cells from future stronger insults and increases their survival [12]. In contrast, cells depleted of HSF1 or one of the heat shock proteins, such as Hsp70, demonstrate attributes of senescence, proving the pivotal role of the chaperone system in the life cycle of the cell [13,14]. Another example of proteostasis mechanism deficiency comes from the study in which aged human fibroblasts lacked their ability to cope with an overproduction of carbonylated proteins because of lowered expression of heat shock proteins [15].

In this article, we briefly overview the function of proteostasis mechanisms in cell aging with an emphasis on molecular chaperones and their interrelationships with autophagy and UPS.

## 2. Functions of Chaperones in Aging Cells

Chaperones, most of which belong to heat shock protein families Hsp100, Hsp90, Hsp70, Hsp60, and small Hsps (such as Hsp27), are expressed in cells affected by a huge variety of stressful factors, drugs and even emotions in all known cells and organisms. Their function, commonly known as the heat shock response (HSR), is regulated by a few heat shock transcription factors (HSFs), which are activated by the release of their molecules from complexes with Hsp90 and Hsp70, phosphorylation of certain amino residues, proteolytic degradation, trimerization, and intranuclear transport of the ready molecule. The correct sequence of the above events remains elusive to date [16,17]. Analyzing the list of HSF modulators, one can conclude that among the signaling pathways involved in HSR activation are cascades based on multipurpose AMP-activated protein kinase and kinases C and A, most of which are known to regulate cell proliferation and aging [17]. Because of the promotion of cell growth by the activation of transcription of hundreds of genes, HSF1 becomes a potent factor contributing to the development of malignancy, as observed in numerous patients with various types of tumors [18], causing the burden of cancer-associated fibroblasts in the tumor microenvironment, as reported by the laboratory of Sue Lindquist [19]. On the other hand, the stress response mechanism is essential for the normal development of a variety of organisms from *C. elegans* to humans [19,20]. The involvement of HSF1 in the regulation of aging is proven by the data, indicating that the factor was found to elevate the transcription of epidermal growth factor (EGF)-containing fibulin-like extracellular matrix protein 1, calcium voltage-gated channel subunit α1A, and a Jun proto-oncogene typical of the aging process in mammalian cells [21]. In *C. elegans*, HSF1 increases both stress tolerance and lifespan and acts in close cooperation with another transcription factor, DAF-16, contributing to the longevity of nematodes [22]. The factor also triggers the expression of several genes important for normal aging in *C. elegans*, including sip-1 (sHsp) and cyp-35B1 from the cytochrome P450 family [23].

The improper functioning of HSF1 in aged cells was first reported in 1998, when it was shown that a decline in Hsp70 synthesis was paralleled by a decrease in the levels of HSF1 in IMR-90 lung cells and in skin fibroblasts derived from old human subjects [24] (Table 1). Earlier, the deficiency in the active transcription of *hsps* genes in response to heat shock or similar stressors that was defined as the reduced level of Hsp70 mRNA was linked to advanced age, and this decrease was found practically in all cells/tissues studied, including the brain, lung, skin, neurons, hepatocytes, macrophages and fibroblasts (Table 1). More recently, detailed investigation of the age-related heat shock response in the brain and heart of mice carried out with the aid of chromatin immunoprecipitation, qPCR, Western blotting, and enzyme immunoassay revealed no difference in chaperone expression between young and old mice in all brain regions. In contrast, the authors observed an age-related reduction of chaperone levels in the heart [25]. Although the number of species whose cells were subjected to a comparative analysis of HSR was limited to about 8–10, most of the data corroborated the lowering of the activity of the chaperone-based proteostasis mechanism in aged cells.

Additionally, the growing body of data demonstrates that knockdown of HSF1 leads to an inappropriate reaction of cells to oxidative stress, one of the most relevant aging factors [26]. Consequently, the depletion of HSF1 in human fibroblasts caused their senescence, which was associated with the activation of the p53-p21 pathway but not related to the downregulation of Hsps [13]. It is noteworthy that VER155008, an inhibitor of Hsp70, increased proteotoxicity and suppressed proliferation, but did not induce senescence, proving that damage of HSF1-regulated proteostasis may not be the only factor causing aging [13]. Since HSF1 has been shown to regulate stress granules, serving as its storage depot, another pathway used by *C. elegans* to reactivate HSF1 is to inhibit the formation of stress granules. The insulin/IGF-1 signaling pathway assists in this action, thereby activating HSF1 in the aging process [27].

In conclusion, aging is associated with the inhibition of HSF1 activity, and this affects the ability of cells to respond to a great variety of harmful factors related to senescence. The negative regulation of HSF1 in most of cells or tissues studied to date occurs in a complicated way and may lead to (i) the inappropriate response of cells to cytotoxic factors (which is typical of aging) and (ii) the modulation of other anti-stress or repair systems, such as autophagy or UPS, as shown in multiple experiments on *C. elegans* [28]. Such interactions are discussed separately in Section 5 of this review.

The expression of chaperones, mostly controlled by HSF1 and the constituting so-called “chaperome”, has been explored in human brain tissue [42]. The bioinformatic analysis of 99 transcriptomes covered 332 chaperone proteins and showed that the expression of 32% of the chaperome, corresponding to ATP-dependent chaperones, was repressed, whereas 19.5%, corresponding to ATP-independent chaperones and co-chaperones, were elevated in aged humans. Interestingly, these categories were even more pronounced in the brains of those with Alzheimer’s, Huntington’s, or Parkinson’s disease [42]. This massive analysis was preceded by decades-long studies in which less advanced techniques were employed to measure basal amounts of certain chaperones or levels of their expression in cells subjected to stressful conditions. Both of these parameters are of value because the basal content or synthesis level of a Hsps means the readiness of a cell or tissue to respond to a certain stressful factor, while the speed of such a reaction is measured as the degree of HSF1 activation and increase in the amount of mRNA or protein. These data for the three major Hsps implied in cellular proteostasis mechanisms are presented in Table 1.

The members of the Hsp70 (HSPA) family are the most abundant proteins, whose synthesis is elevated as a result of HSF1 activation [43]. These proteins are known to recognize newly-synthesized or damaged polypeptides and to refold their molecules, or to direct incorrigible structures to ubiquitination and finally to proteasomal degradation. In this cycle, co-chaperones belonging to the DNAJ family expose the substrate polypeptide to the major chaperone (Hsp70 or Hsc70 constitutive member), with the simultaneous switching of ATPase activity [44] (Figure 1). Co-chaperones of Hsp110 or BCL-2-associated athanogene (Bag) families, both serving as nucleotide exchange factors, dissociate ATP from the Hsp70 molecule and prepare the latter for the next cycle of chaperoning [45]. The polypeptides with irreversibly-damaged structures are channeled to the proteasome through the complexes of Hsp70 or Hsc70 with C-terminal Hsc70 interacting protein (CHIP) (Figure 1, see Chapter 4 for more details). In a single mammalian cell, Hsp70 or Hsc70 compose a few of the supramolecular structures, such as the cystic fibrosis transmembrane conductance receptor, clathrin transport system, and others; moreover, Hsc70 plays a significant role in the assembly of CMA lysosomes, as well as in numerous glucocorticoid receptors, together with Hsp90 and their co-chaperone HOP [46] (Figure 1).

Hsp70 has yielded mixed results that give no indication of a common or consistent function as it relates to aging. For instance, in human fibroblasts, Hsp70 content tends to increase with a number of passages (or aging), and this fact is explained by the fact that cells transitioning to the senescence face much more stressful factors than the younger ones [33]. Similar elevation of Hsp70 content in human leukocytes and lymphocytes derived from aged persons is thought to be linked to the enhanced degree of cell differentiation [36]. Controversial results were also obtained in the analysis of Hsp70 content in rat skeletal muscle. First, it was found that aging is accompanied by a reduction of Hsp70, along with the weakening of contractile activity [37,47]. The other group demonstrated an increase in the basal level of Hsp70 in the muscle of older animals, and they explained this fact by a functional relationship between the need of the chaperonic activity of Hsp70 and of its protective power against the age-related apoptosis [36]. Together, these data indicate that aging is often associated with the reduced production of a major cytoprotective Hsp70 molecule, and in some cases, this deficiency may be compensated by triggering based on the HSF1 transcription mechanism (see Table 1).

The molecular chaperones of the Hsp90 family play an important role in proteostasis, the regulation of metabolism, and supporting oncogenic (client) proteins in latent form. A list of the latter includes HSF1, whose release from the complex with the chaperone activates the synthesis of corresponding genes [48]. The functional activity of Hsp90 is dependent upon ATP, so that, in case of ATP depletion, client proteins undergo ubiquitin-mediated proteasomal cleavage. As a result, the processes they control get inhibited. Since most Hsp90 client proteins are important molecules for oncogenic processes, tens of Hsp90 inhibitors are developed currently as anti-cancer drugs. Interestingly, a few of Hsp90 inhibitors, due to their ability to release HSF1 and to activate it, are employed, while in laboratory research, as inducers of Hsp70 [49].

The works on Hsp90 mRNA and protein content in senescent cells are sparse as compared to that for Hsp70 and show both the reduction of the protein concentration and activity in aging tissues. According to Nardai and co-authors, the chaperonic activity of hepatic Hsp90 in older rats was much lower than in younger animals (Table 1) [38]. The observed decrease in chaperone capacity may be explained by Hsp90 overload due to the increased amount of damaged hepatic polypeptides or could reflect the direct proteotoxic damage of the chaperone. Similarly, a strong reduction in Hsp90 content was found in late passage senescent human fibroblasts [33]. In contrast, an increase in Hsp90 immunoreactivity was found in the hippocampus of older gerbils compared with that of adult animals [39]. We suggest that the low content of Hsp90 may be beneficial for the healthy aging of a normal cell, and this suggestion is in line with the recent discovery of senolytics based on inhibitors of Hsp90 [50].

Senolytics are substances that improve quality of life and extend life expectancy. In their search, one group used embryonic fibroblasts with reduced DNA repair capacity, which senesce rapidly if grown, even at atmospheric oxygen. These fibroblasts were treated with a number of various substances, and 17-DMAG, an inhibitor of Hsp90, was found to decrease the activity of β-galactosidase in the appropriate assay. Treatment of mice, imitating a human progeroid syndrome, with this Hsp90 inhibitor extended lifespan, delayed the onset of age-related symptoms and reduced the expression of p16INK4a (a marker of senescence) [50]. Investigators of the other group, using age-stratified human transcriptomes, created age classifiers and applied them to transcriptomic changes induced by >1000 compounds, arranging the above compounds by their ability to induce a “youthful” transcriptional state in human cells [51]. Two compounds, monorden and tanespimycin (both inhibitors of Hsp90), were reported to extend the lifespan of *C. elegans* and improve its physiology under proteotoxic stress. These beneficial effects on nematodes were attributed to the HSR mechanism acting via HSF1 which is permanently blocked by Hsp90.

Hsp27 (HSPB1) belongs to a group of ten sHsps, whose expression is triggered by many environmental and pathophysiological stressors that cause damage to the structure of cellular proteins. The chaperone activity of Hsp27, in contrast to Hsp70 and Hsp90, is ATP-independent and induced by phosphorylation of the protein and its oligomerization [52]. It has been proven that the sHsps interact with monomers or oligomers of damaged stress polypeptides or amyloidogenic proteins, preventing their aggregation and/or promoting their proteolytic degradation. This action can be performed by Hsp27 alone or in combination with other Hsps. The sHsps are multi-functional proteins, and among their major activities is the inhibition of apoptosis and rearrangement of the actin cytoskeleton [53]. Hsp27 possesses antioxidant activity and shows high cytoprotective potential in numerous models of human disease, and these two attributes impact on its implication in oncogenesis. Since the chaperone’s expression becomes highly upregulated after chemotherapy, its content is related to the bad prognosis of many cancers [54]. Aging, as the process of accumulation of cytotoxic factors (such as oxidative stress), in all subjects tested to date has been caused by an increase of the sHsps (see Table 1). This trend was especially pronounced in skeletal muscle permanently functioning in conditions of increased levels of ROS [40,41,55]. Greatly-enhanced production of Hsp25 was also found in all parts of the brain of aged rats, which again can be explained by the neuron’s need for protection from oxidative stress [56]. It is noteworthy that, in tissues taken from the older subjects, sHsps were found preferably in phosphorylated form, which proves that the cells respond to the damaging factors of aging not only by enhanced expression of the chaperone but also by its activation. The activation of the major sHsps Hsp27 and αB-crystallin is triggered by a complex kinase cascade, including p38 mitogen-activated protein kinase (MAPK), MAPK-activated protein kinase-2, and extracellular signal-regulated kinase-1/2. All these enzymes were found in the muscles of aged animals in the phosphorylated state, and the amount of this form was higher compared with the muscle of younger counterparts [55].

The process of age-related pathologies can also be associated with the state of particular sHsps. To illustrate such a link, one can turn to myopathies caused by the mutation of R12OG in the αB-crystallin gene belonging to the small Hsp family. The mutated gene was overexpressed in cells of different origins, and this was found to induce the pathogenic aggregation of desmin, which is known to be a cause of this type of myopathy. Based on their data, authors emphasize that some cellular factors (or situations) can influence the course of pathogenesis in muscle, and this could explain the late-onset characteristics [57]. Another study focused on the model of cardiomyopathy based on transgenic mice expressing the αB-crystallin gene with R120G mutation. This transgene caused reductive stress in heart tissue accompanied by reduced glutathione, which was due to the increased expression and enzymatic activity of glucose-6-phosphate dehydrogenase (G6PD). The intercross of cardiomyopathic animals with reduced G6PD levels rescues the progeny from cardiac hypertrophy and protein aggregation [58].

Together, the data presented in this section lead us to some conclusions. First, the basal amount of major chaperones, Hsp70, Hsp90, and sHsps, in aging cells is strongly dependent on a tissue or cell type, on their metabolic state, and on stresses that are disposed of during the period of aging. In most of the probes/subjects tested, the chaperones carry out pro-survival functions, which are good for a healthy person and bad for a cancer patient. Secondly, the machinery of the HSR loses its activity in aging cells, but in rather rare cases, it may be reinforced by synthetic or natural compounds or by training, as in the case of skeletal muscle. Thirdly, the chaperonic part of the proteostasis system does not function alone in stressful or pathophysiological conditions. It is supplemented with the mechanisms used to clear off the unnecessary protein material. The remaining parts of the manuscript will be dedicated to these mechanisms.

## 3. Autophagy is a Process of Cell Upgrading

Autophagy is an evolutionarily conservative catabolic process that is activated in response to extra- and intracellular stressors and is generally established to support cell viability by recycling long-living proteins and damaged organelles. At the same time, intensification of autophagy can lead to programmed cell death, apoptosis. Four types of autophagy are currently distinguished: macroautophagy, microautophagy, selective, and CMA.

Microautophagy occurs when fragments of cell membranes and macromolecules are simply captured by the lysosome (Figure 1). Macroautophagy is a more complex process and happens when part of the cytoplasm and damaged organelles are surrounded by a membrane compartment [59]. With CMA, directed transport of partially denatured proteins from the cytoplasm to the lysosome cavity occurs. Selective autophagy is determined by the modification of substrates, for example, ubiquitination [59]. In our review, we paid the most attention to macroautophagy and CMA since these two mechanisms are most associated with and chaperones and their relevance in cell aging is better enlightened in the literature.

Macroautophagy is tightly-regulated by multiple signaling pathways based on specific proteins whose functions have not been fully elucidated to date [60]. “Autophagy-related genes” (Atgs) were first discovered in yeast cells, and later their analogs were found in mammals [61]. The products of these genes initiate autophagy and control the maturation of autophagosomes. The main regulator of autophagy, mammalian target of rapamycin complex (mTOR), suppresses autophagy by phosphorylating and inhibiting the key upstream serine-threonine kinase ULK1 that activates macroautophagy [62]. The process of macroautophagy consists of several stages: (i) initiation, (ii) nucleation, (iii) elongation, (iv) maturation, (v) fusion and (vi) degradation [63].

For autophagosome formation, a number of proteins are recruited, including key players such as Atg5, Atg14, Atg12, Atg9, Atg16L, Vps34, Beclin1, Ambra, and LC3-II, which are gradually included in the processes [64,65,66,67]. LC3-II was found on both the inner and outer membranes of the autophagosome, where it participates in the selection of cargo for degradation. Cargo transport is regulated, on the one hand, by special adapter proteins, including the p62/SQSTM1 protein, and on the other hand, by LC3-II that acts as a receptor of protein aggregates or organelles. The autophagy cycle ends at the fusion of the autophagosome with the lysosome, resulting in both the cleavage of cargo proteins and of p62/SQSTM1 [65]

Mutations of the Atg1, Atg7, Atg18, and Beclin1 genes have been shown to shorten the lifespan of *C. elegans* [68], and a reduction in the expression of the Atg1 and Atg8 genes decreases the longevity of *Drosophila melanogaster* [69]. Knockout of Atg genes in the mouse model leads to the death of mice in the postnatal period, which indicates that Atg genes perform a certain function during organogenesis and differentiation [70].

The inhibition of macroautophagy induces aging, which is accompanied by various pathologies. The restoration of macroautophagy abolishes aging and increases life expectancy, but this is also accompanied by a significant increase in the number of spontaneous tumors in mice [71], perhaps because genomic instability increases in old cells [72]. Macroautophagy provides genetic integrity by increasing homologous recombination, increasing excision repair, and participating in the DNA damage response [73].

Chaperone-mediated autophagy (CMA) links two components of cell proteostasis: molecular chaperones and autophagy. CMA is a more selective mechanism of unappropriated protein degradation, and it was identified predominantly in mammals and birds [74]. Using this mechanism, approximately 30% of cellular proteins could be cleared, including both long-living proteins (such as glyceraldehyde-3-phosphate dehydrogenase) and short-living proteins (such as HIF1α) [75].

The most important actors in this process are Hsc70 (HSPA8) as chaperone and lysosome-associated membrane glycoprotein LAMP2A, which is a lysosomal surface receptor [76]. The selectivity of the process is provided by only proteins containing the peptide with the sequence similar to KFERQ (which is the substrate for Hsc70) and are targeted by CMA [75]. Hsc70 is able to transfer the target polypeptide with the KFERQ-like motif to the cytosolic C-terminal tail of LAMP2A, triggering its assembly to a multimeric translocation complex through which the substrate protein reaches the lysosomal lumen for degradation. The first group includes monomers of the glial fibrillary acidic protein (GFAP, a component of intermediate filaments) and its GTP-binding partner, elongation factor 1α (EF1α). Both were found to have an impact on the assembly of the LAMP2A complex [77]. Another group consists of the kinases mTORC2 and AKT1 and phosphatase PHLPP1 [78]. mTORC2, the resident of the lysosomal membrane, phosphorylates AKT1, whereas PHLPP1 causes its dephosphorylation. Non-phosphorylated GFAP favors the formation of high molecular weight complexes of LAMP2A [79], but being phosphorylated by ATK1, pGFAP represses CMA activation, inhibiting the assembly of the LAMP2A assembly [80]. Lysosomal Hsc70, together with Hsp90, also contributes to the assembly and disassembly of LAMP2A on the lysosomal membrane [77]. Further disassembly or degradation of LAMP2A occurs in the lipid microdomain of the lysosomal membrane with cathepsin A and metalloproteinases.

The activity of CMA is shown to vary depending on the tissue and even inside a single organ. A striking zonal difference in the liver was noted, as was recently demonstrated with the KFERQ-Dendra mouse model [81], and CMA activity should be modified with age. Many proteotoxic pathologies like Alzheimer’s disease and Parkinson disease usually appear in late age. At least in part, this phenomenon is related to the malfunction of the degradation systems that cannot prevent the formation of aggregates of improperly-folded proteins [82]. Reduction of CMA efficacy occurs in normal physiological aging, as it was demonstrated in old rodent and in cellular models [83,84], and results in a decrease of LAMP2A expression, limiting its ability to bind substrate and capacity to uptake the substrate [85]. However, in aged cells, the lipid context of the lysosomal membrane differs from that of young cells and could alter the stability and dynamic of LAMP2A in lysosomes [86], which generally leads to disruption of CMA mechanisms in old cells and a decrease of intracellular protein degradation in general.

## 4. UPS is Necessary for Correct Aging

The ubiquitin-proteasome system (UPS) is another principal pathway for protein degradation in the eukaryotic cell. The process involves the sequential conjugation of ubiquitin moieties to proteins (ubiquitination), which is catalyzed by an enzymatic cascade composed of ubiquitin-activating enzymes (E1), ubiquitin-conjugating enzymes (E2), and ubiquitin ligases (E3) (Figure 2). At first, the ubiquitin is activated by its conjugation to E1 in an ATP-dependent manner. Then, the activated ubiquitin is transferred to E2, which associates with an E3 ligase that promotes the transfer of the ubiquitin moiety to the target protein (Figure 2). The human genome encodes for two E1s, about 40 E2s, and more than 800 E3s [87]. These E3s are largely responsible for substrate specificity and can be divided into four major families depending on their ubiquitination domains: (1) the really interesting new gene (RING), (2) the homologous to E6-AP carboxyl terminus (HECT), (3) the U-box, and (4) RING-IBR-RING (RBR) E3 ligases [88]. Once the first ubiquitin is attached to the substrate, E3 can elongate the ubiquitin chain by creating an isopeptide bond between any of its seven Lys residues (6, 11, 27, 29, 33, 48, 63) [89]. In addition to the lysine side chains, the N-terminal Met1 is a donor for additional ubiquitin attachment, forming linear polyubiquitin chains [90]. Thereover, polyubiquitin chains of different structural topology can be generated, and these different chain types guide the substrate protein to specific signaling pathways, including degradation by the proteasome, endocytic trafficking, inflammation, translation, and DNA repair. The conjugation of ubiquitin moieties to substrates is reversible, and the former can be detached and adjusted by deubiquitination enzymes (DUBs) (Figure 2). The human genome encodes approximately 100 DUBs [91]. Many DUBs are associated with the proteasome, where DUB-mediated removal of ubiquitin from substrates is important for ubiquitin recycling.

One of the E3 ligases, CHIP, plays an important role in cellular proteostasis by binding to Hsp70 and Hsp90 and providing the ubiquitination and proteasomal degradation of polypeptides that are targeted by the chaperones [92]. The proteins targeted by CHIP are involved in multiple signaling pathways, and its dysregulation may lead to pathologic conditions [93]. Among CHIP’s targets, there is the conserved insulin/IGF-1 signaling pathway, genetically associated with longevity across species [94]. In *C. elegans*, CHN-1, a CHIP nematode analog, can directly ubiquitinate the insulin receptor DAF-2. Activation of DAF-2 leads to the initiation of PI3/AKT, which phosphorylates FOXO (a transcription factor), preventing its accumulation in the nucleus. Reducing DAF-2 signaling through ubiquitination allows DAF-16 to accumulate in the nucleus and activates the genes responsible for longevity [95].

Recent findings suggest that proteasome function declines during aging at different levels, including decreased expression of proteasome subunits [96], reduction of proteasomal activity [97], alteration and/or replacement of proteasome subunits [98], and disassembly of proteasomes [99] or their inactivation by interacting with protein aggregates [100]. Importantly, proteasome inactivation by aggregates can lead to a negative feedback loop during aging, because it may induce the avalanche-like accumulation of misfolded proteins and formation of new inclusions, which in turn can further inhibit proteasome activity. An example is the accumulation of protein inclusions in yeast caused by the failure of the UPS during aging [101]. Interestingly, the disaggregation restored proteasomal degradation of the substrate in aged cells without elevating proteasome levels, demonstrating that age-associated aggregation obstructs UPS function. In general, a decline in proteasome function during senescence and aging has been observed in several mammalian tissues and cells including human muscle [102,103] lens [104], lymphocytes [105], keratinocytes [106] and fibroblasts [97,107], rat heart [108], spinal cord [109], lung [110], liver [111], retina [112], brain [113] and muscle, mouse brain [114,115], spinal cord [116] and adipose tissue [117].

Indeed, partial inhibition of proteasome activity induced a senescence-like phenotype of early passage human fibroblast cells [118] and, on the other hand, overexpression of the β5 catalytic subunit in primary human fibroblast cell lines contributed to elevated levels of other β-subunits and increased proteolytic activities, resulting in the extension of the lifespan [119]. Since inhibition of proteasome activity is known to induce rapid cell death through apoptosis, stimulating the proteasomal activity could be an effective strategy to treat age-related disorders.

The mechanisms of upregulating proteasome activity in mammalian cells are less defined. There are several approaches used to increase the activity of proteasomes which are based on increased expression of proteasomal proteins, the activity of proteasome assembly chaperones, modulation of 26S proteasome assembly and stability, induction of conformational alteration leading to the opening of the proteasome gate, and stimulation by small molecules through allosteric interactions that enhance substrate binding and/or degradation in one or more catalytic sites [120,121]. Exposure to environmental stresses that activate HSF1 could also elevate the activity of proteasome subunits in mammalian cells [122] and increase the proteasomal activity as a whole [123]. Conversely, the expression of activated HSF1 in mouse embryonic fibroblasts and human myoblasts upregulated Hsps but not affected expression and activity of proteasomes [124]. On the other hand, HSF2-deficient mouse embryonic fibroblasts exhibited a reduced proteasome activity [125]. Interestingly, in mild heat stress conditions, proteasome activity was induced when Hsp70 expression is depleted in human primary fibroblasts [126]. Moreover, Hsp70 plays the role of the activation of the proteasome through dissociation and subsequent reassociation of the 26S proteasome during adaptation to oxidative stress [127]. In addition, Hsp90 was shown to play a principal role in the assembly and maintenance of the 26S proteasome, because the functional loss of Hsp90 caused dissociation of the 26S proteasome [128]. In summary, although a few chaperones that could affect proteasomal activity have been identified, universal mechanisms underlying the regulation of proteasome expression and activity remain to be elucidated.

## 5. All Three Proteastasis Systems Cooperate In Aging

In this section, we overview the pathways linking the two proteolytic machineries of proteostasis to chaperones in an attempt to understand their common role in aging. One of the most important elements in the regulation of response to damaged proteins, HSF1, controls, besides Hsps, the expression of p62 and Atg7, whose products are typical of autophagy mechanisms [129,130]. p62 was found to extend lifespan and improve proteostasis in *D. melanogaster* and *C. elegans* models in an autophagy-dependent manner, and its enhanced synthesis induced by hormetic heat shock in *C. elegans* improved the fitness of mutants with defects in proteostasis [131,132]. Based on these data, we hypothesize that the reduction of p62 synthesis due to the reduced ability to activate HSF1 (see Table 1) may result in deficiency of correct autophagy observed in aged cells. Another autophagy regulator, Atg7, whose synthesis is also controlled by HSF1, is indispensable for aging cells, since its dysfunction causes the decreased lifespan in *C. elegans* [68]. The knockout of Atg7 in mice induced age-related phenomena, such as protein aggregation and neurodegeneration [133], increased protein oxidation [134], and disordered and defective mitochondria [135] in younger animals.

A key modulator of cell aging is mTOR kinase, which controls (besides autophagy) actin cytoskeleton organization, metabolism, and also serves as a sensor of the level of energy and nutrients in a cell [136]. It has been shown that mTORC1 can be inhibited by c-Jun N-terminal kinase as a result of proteotoxic stress. Conversely, HSF1 preserved mTORC1 integrity and function by inactivating JNK, irrespective of its transcriptional activity [137]. While it is not yet clear which mechanism, based on HSF1 or mTORC, plays the primary role in age-related disbalance of proteostasis, the data of Su et al. show that chaperonic and autophagy systems are well coordinated and the fall of the former should lead to the upregulation of autophagy as a compensatory action.

A similar compensatory mechanism has been demonstrated in cross-talk between the AMPK-based system of metabolic stress sensing and HSF1 transcription mechanism. First, metformin (chemical activator of AMPK, metabolic stressor and popular anti-diabetic drug) was found to induce AMPK-dependent phosphorylation of HSF1 at Ser^121^, which inactivated the factor and provoked proteotoxic stress in tumor cells [138]. Thus, the authors established a direct link between the activation of the metabolic stress mechanism and diminished response to proteotoxicity. A number of reports demonstrated that AMPK phosphorylates the autophagy-related proteins belonging to mTORC1, ULK1, and PIK3C3/VPS34 complexes and activates autophagy (see Reference [139] for review) (Figure 3). Importantly, in senescent cells, AMPK was inactivated, while the pharmacological activation by metformin or berberine prevented the development of a senescent state and reduced the deleterious effect of hydrogen peroxide on autophagy efficacy, as proved by the enhanced level of p62 degradation and hydrolase activity [140]. It is noteworthy that metformin, due to its ability to induce autophagy by activation of AMPK, is considered a potential hormesis-inducing factor with geroprotective properties [141].

Regulatory links between the UPS and chaperones are also well-documented. One such link is provided by CHIP, which was found to serve as an E3 ubiquitin ligase that targets misfolded proteins for their degradation through the 26S proteasome. The protein was also found to assist major chaperones in the folding of selected polypeptides, suggesting its role in sorting out of needless polypeptides [142]. Mechanistically, CHIP links proteasomes with Hsp70 or Hsc70 chaperones through nucleotide exchanging factors that are Bag domain-containing proteins (e.g., Bag-1, Bag-3, etc.) [143]. Of note, besides its function in UPS-chaperone mechanism linking, CHIP was found to regulate trimerization and transcriptional activation of HSF1 by remodeling the complex of HSF1 with Hsp70 and Hsp90, affecting the latter conformation [144]. This evidence of CHIP-related cytoprotective activity has been confirmed by data indicating that STUB1 (alternative name of CHIP) ubiquitinates and targets the proteasome ARNT-like 1 (BMAL1), also called aryl hydrocarbon receptor nuclear translocator-like protein [145]. As a result of BMAIL1 proteasomal degradation, the senescence-inducing effect of hydrogen hydrochloride was attenuated, proving the high pro-survival potential of CHIP/STUB1 in an oxidative stress-based model of senescence. Similarly, STUB1 was shown to be subject to degradation by phosphorylated TFEB, a regulator for transcription of genes involved in autophagy and lysosome biogenesis; this effect was corroborated with increased autophagosome and mitochondria biogenesis [146]. These data allow us to present CHIP/STUB1 as a factor linking not only chaperones with the UPS but also the latter system with the autophagy/mitophagy mechanism.

The role of CHIP/STUB1 in activating autophagy was convincingly demonstrated in several studies, featuring one in which cells with a reduced level of the protein were employed to analyze their response to bafilomycin A, an inhibitor of autophagy. In these cells, knockdown of CHIP caused a decrease in AKT/mTOR activity and reduction of ULK1 phosphorylation on Ser^757^. Since the degradation of p62 was inhibited, the authors concluded that CHIP knockdown affected autophagic flux [147]. The data obtained on CHIP -/- transgenic mice modeling pressure overload with the application of fenofibrate, an antagonist of peroxisome proliferator-activated receptor alpha (PPARα) showed that the full abolishment of CHIP caused an increase of cardiac fibrosis accompanied by reduced myocardial function. Importantly, the most vulnerable targets for fenofibrate in CHIP knockout mice were mitophagy and autophagy mechanisms [148]. The importance of CHIP for the survival of neural cells subjected to neurodegenerative pathogens was proved in studies also performed on CHIP -/- mice. The results demonstrated that CHIP deficiency was accompanied by a failure in the clearance of mitochondria damaged by stress, along with impaired energetic and redox homeostasis [149].

In its linking activities, CHIP is accompanied by Bag proteins specifically interacting with the ATPase region of Hsp70 via their conserved C-terminal Bag domains [150,151]. Two of them, Bag-1 and Bag-3, are of particular interest because of their functional diversity in cell physiology. Under normal conditions, a high Bag1- expression, but a low Bag-3 expression, is observed, while under pathophysiological conditions, the Bag-3 level is elevated, and the Bag-1 level is decreased. The transition from Bag-1 to Bag-3 expression is accompanied by a functional switch from Hsp70-Bag-1-mediated proteasomal degradation to Hsp70-Bag-3-mediated selective macroautophagy [152]. In aging, IMR90 cells and Bag1 and Bag3 expression levels were regulated reciprocally, depending on the number of passages (aging) and acute stress; so, in comparison to young fibroblasts, an increased expression of Bag3 and reduced expression of Bag1 was found in aged fibroblasts [153]. In addition to an increased Bag3 expression, aged or stressed fibroblasts demonstrated an elevated autophagic flux and upregulated levels of the polyUb- and LC3-binding protein p62 and of the early autophagosome marker WIPI [153].

These data are corroborated with the results of McCormic et al., who found that the expression of key molecules involved in the regulation of three tightly-connected proteostasis mechanisms changed with age. A comparison of the transcription profiles of human blood mononuclear cells of young and old donors showed that the expression of p62 and LC3-II in old cells was significantly reduced, and the expression of LC3-I and HSP70, on the contrary, was increased in cells of old donors.

## 6. Conclusions

Cell proteostasis machinery consists of closely-associated parts, and some of them are regulated in a tightly-coordinated manner, especially in response to stressful factors. This coordination is mostly performed by HSF1, AMPK, and mTOR, and additional aging pathways, such as the one based on TFEB or p21waf-p53 axis, can interfere with the former three signaling systems to result in a certain balance in cell homeostasis. We think it is of importance that chaperones, autophagy, and UPS may compensate for each other, making the pattern of homeostasis plasticity more focused for a response to a specific age-related factor. This may explain the variability of expression dynamics for many protein components of the three proteostasis mechanisms in aging or senescent cells. We mentioned a few of the small molecules, including senolytics, AMPK activators (metformin and AICAR), and Hsp70 synthesis inducers, that may activate certain mechanisms of response to age-related pathology and/or aging, and therefore can be applied to generate appropriate cytoprotective power. It is also of note that there are proteins, for example, CHIP/STUB1 and Bag-domain proteins, acting in a manner of molecular hubs, and whose interplay controls the fate of proteins that have become unnecessary to a cell and represent a building material for the synthesis of new biomolecules. In this view, polyvalent molecules like CHIP, Bag(-s), and Ulk1 and their interactions may become prospective targets for the reduction of proteotoxic effect of age-related factors, misfolded protein aggregation, mistakes in intracellular transport, improper function of supramolecular structures, and other signs of stress and aging. Lastly, the above modulators of proteostasis pathways may be applied in combination with classic geroprotectors such as vitamins and antioxidants, which can be effective in extending longevity or at least in improving quality of life.

## Figures and Tables

**Figure 1 cells-09-01308-f001:**
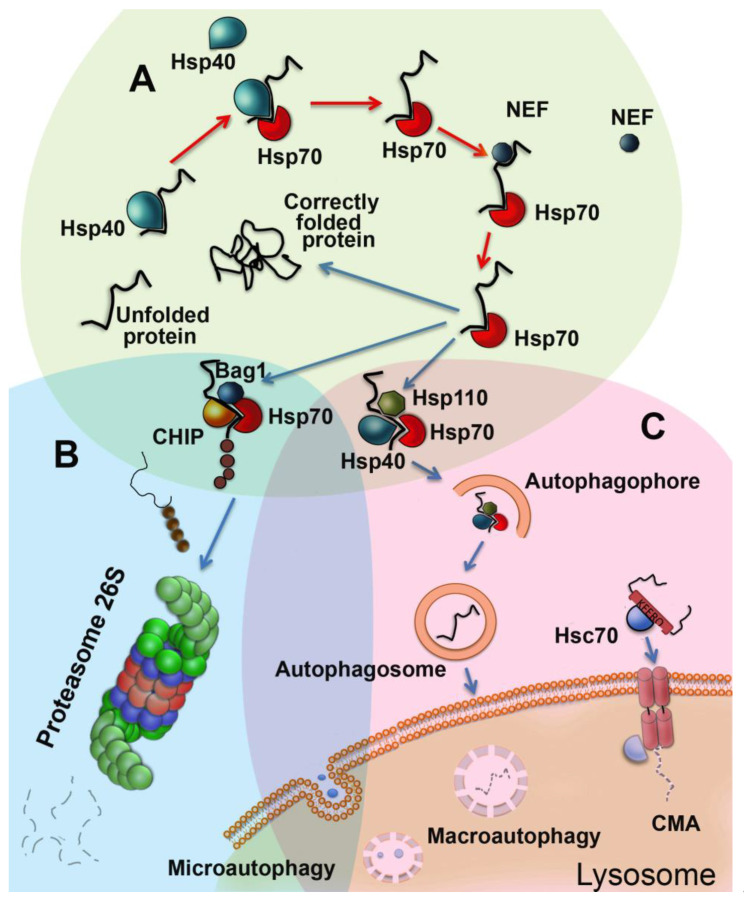
Proteostasis pathways acting in eukaryotic cells in brief. (**A**) Chaperonic machinery based on Hsp70 family members includes, besides Hsp70 itself, Hsp40-like (DNAJ family) proteins and nucleotide exchange factors (NEF, Bag domain-containing, Hsp110). In a chaperonic cycle, Hsp40 exposes a molecule of a newly-synthesized or damaged polypeptide to Hsp70 and concomitantly enhances its ATP-ase activity. Corrected substrate protein is released due to the conversion of Hsp70 molecule from ADP to ATP-bound form, performed with the aid of NEFs. If the substrate is incorrigible it is targeted via Bag-1-mediated recruitment of E3 ubiquitin ligase CHIP for further proteasomal degradation. The Hsp70-Hsp110-Hsp40 complex may also target improperly-structured polypeptides to maturating autophagosomes. (**B)** In the UPS cycle, 26S proteasome obtains and cleaves polyubiquitinated proteins by a variety of ubiquitin ligases, resulting in the production of short peptides (see Figure 2 for further details). (**C**) Autophagy features at least three distinct protein degradation systems, including macroautophagy, which serves for digestion of polypeptides and organelles, microautophagy necessary for the degradation of useless membranous structures, and chaperone-assisted autophagy which targets KFERQ motif-exposing proteins. All types of autophagy possess powerful hydrolytic activity digesting substrates down to amino-acid and supplying cells with nutrients.

**Figure 2 cells-09-01308-f002:**
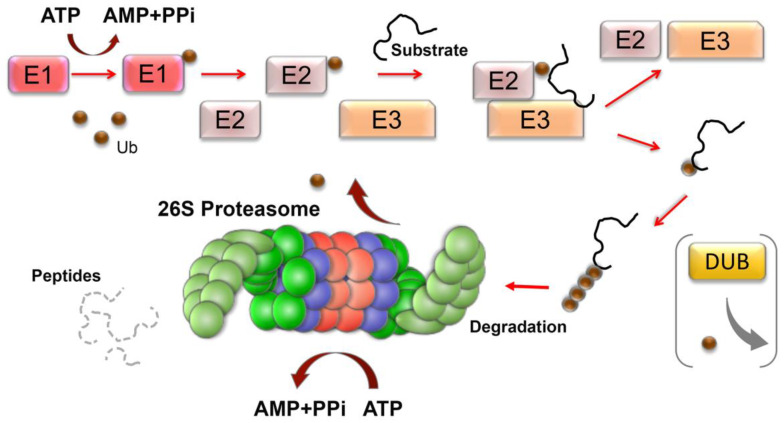
The ubiquitin-proteasome system of cell proteostasis in brief. The ubiquitylation system consists of activating enzymes (E1s), conjugating enzymes (E2s), and ligase enzymes (E3s) that result in the activation and conjugation of the 76 amino-acid ubiquitin (Ub) onto the lysine residues of targeted proteins. After the first ubiquitin has been attached, the E3 can elongate the ubiquitin chain by creating ubiquitin–ubiquitin isopeptide bonds. Ubiquitination can be reversed from the substrate by deubiquitinating enzymes (DUBs). Polyubiquitinated proteins are recognized and degraded by the 26S proteasome with the release of short peptides and reusable ubiquitin moieties.

**Figure 3 cells-09-01308-f003:**
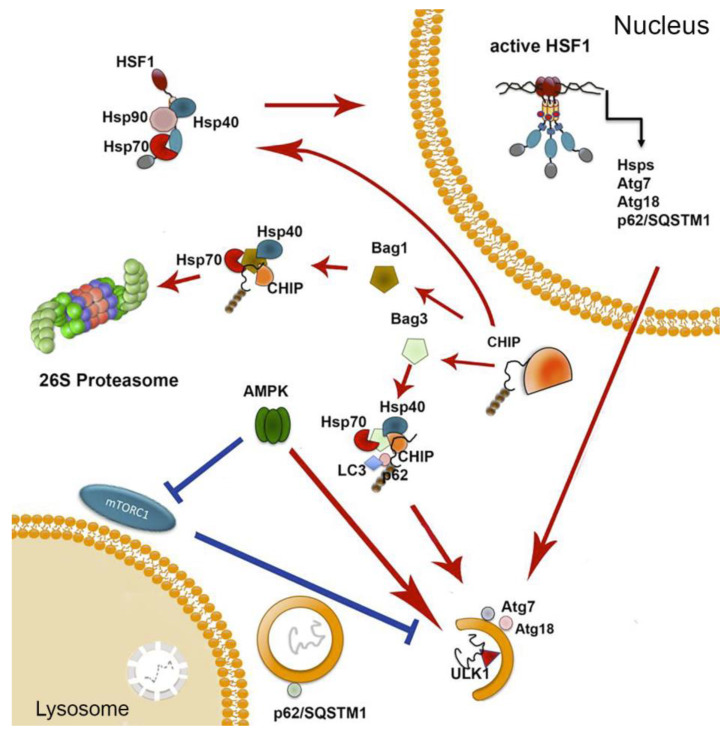
Functional links between proteostasis machineries. The major modulator of heat shock response HSF1, besides its classic downstream targets (Hsps), regulates the expression of proteins related to autophagy function: p62, substrate, and indicators of autophagosomes, as well as Atg7 and Atg18 participating in autophagosome assembly; all three proteins are implicated in the aging process. HSF1 was also shown to protect mTORC1 from JNK-mediated inactivation. AMPK was implicated in the regulation of both mechanisms: the activation of autophagy through phosphorylation of ULK1 and mTORC1, as well as inhibition of HSF1, also by the factor’s specific phosphorylation. The CHIP/Bag-1/Hsp70 complex is able to target a protein substrate to proteasomal degradation; cooperation of CHIP with Bag-3, LC-3, and p62 leads to a substrate targeting to autophagy. mTORC1 and AMPK can phosphorylate ULK1, but with distinct results. Depending on caloric supply, mTORC1 may phosphorylate ULK1 at Ser^757^, and via this inhibit autophagy; alternatively, in the absence of energy, AMPK phosphorylates ULK1 at Ser^555^, leading to the stimulation of autophagy.

**Table 1 cells-09-01308-t001:** Expression of chaperones in aging and stressed cells.

Subject	Normal Conditions						Response to Stress						Ref
	HSF1		mRNA		Protein		HSF1		mRNA		Protein		
The Whole HSR	Y	O	Y	O	Y	O	Y	O	Y	O	Y	O	
Mouse brain cells							+	+	+	+	+	+	[26]
Mouse myocardium							++	+	++	+	++	+	[26]
Hsp70													
Rat brain, lung, skin									++	+			[29]
Rat neuron									++	+			[30]
Rat hepatocytes							++	+	++	+	++	+	[31]
Rat macrophages							++	+	++	+	++	+	[32]
IMR-90 cells, human fibroblasts							++	+	++	+	++	+	[25]
Human fibroblasts (senescence)					+	++							[33]
Human dermal fibroblasts					++	+	+	+	+	+	++	+	[34]
Human monocytes, lymphocytes									++	+			[35]
Rat skeletal muscle					+	++							[36]
Rat muscle							++	+			++	+	[37]
Hsp90													
Human fibroblasts					++	+							[33]
Rat liver (chaperone activity)					++	+							[38]
Gerbil brain cells					+	++							[39]
Hsp27 (small Hsp family)													
Human fibroblasts					+	++							[33]
Rat skeletal muscle					+	++							[36]
Rat muscle					+	++							[40]
Hsp20 & Hsp22 (small Hsp family)													
Gerbil brain cells					+	++							[39]
Rat muscle					+	++							[41]

Y—young; O—old; +—low level of parameter; ++—high level of parameter.

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
