# Peer review of "Molecular Chaperones and Proteolytic Machineries Regulate Protein Homeostasis in Aging Cells"

_cells, 2020, doi:10.3390/cells9051308_

Round 1

Reviewer 1 Report

This review describes the function of proteostasis mechanisms in aging cells. The topic is in line with the interest of Cells Journal. The References are appropriate and recent. English language and style are food. So I accept this review in present form.

Author Response

We are coming back with revised manuscript titled “Molecular chaperones and proteolytolitic machinery regulate protein homeostasis in aging cells” for further evaluation in Cells. We are thankful to reviewers for deep analysis of the manuscript. We carefully studied their comments and following the recommendations of Reviewers 2 and 3 we reformatted Table 1, corrected mistakes in the text and added a new part of Conclusions.  We believe that quality of the paper has benefited in result of the response to the comments.

All suggested modifications are highlighted in red in new text version. 

Reviewer 1:

We thank the reviewer 1 for high evaluation of our manuscript.

Reviewer 2 Report

Review summarizes data about molecular chaperones and proteolytic machineries involved in regulation of protein homeostasis in aging cells. From this point is the review focused on an actual and important topic. 

I have some questions and comments:

1. You wrote (lines 76-77): “In this article, we briefly overview the function of proteostasis mechanisms in cell aging with an emphasis on molecular chaperones and their interrelationships with the other two systems.”

The expression “with other two systems” should be described here exactly.  

2. You wrote (lines 92-93): “On the other hand, the stress response mechanism is essential for the normal development of a variety of organisms from C. elegans to mice.”

Do you mean that stress response mechanism is essential for the normal development only to mice? Is it not the truth for human?

3. You wrote (lines 79-80):“Chaperones, most of which belong to heat shock protein families Hsp100, Hsp90, Hsp70, Hsp60 79 and Hsp27 (small Hsps [sHsps]), are expressed …”

Hsp27 is a member of family of small Hsps. From this point is more correct to write families of  Hsp100, Hsp90, Hsp70, Hsp60 79 and small Hsps (such as Hsp27).

4. Table 1 – the informations presented in this Table are not always clear. Table should sumarize information (data) about changes in expression of chaperones in aging and stressed cells. However, in my opininon is the structure of presented Table 1 not ideal and presented data are not complete. Some comments to this table: 

a) It is not defined what exactly means in your Table + and ++.

b) As an example for my comment I choosed Hsp70 and data presented for rat brain, lung, skin (comment could be applied also for another Hsps and cells): For situation Y (young) is in response to stress for mRNA ++, for O (old) is for mRNA +. This indicates decline in mRNA levels. Were these changes in mRNA associated with protein levels modification? Are not available data about the Hsp70 expression at normal conditions?

5. Legend to the Figure 1.

There is sentence (lines 137-139): “Autophagy features at least four distinct protein degradation systems including macroautophagy that serves for digestion of polypeptides and organelles, microautophagy necessary for the degradation of useless membranous structures, and chaperone-assisted autophagy which targets KFERQ motif-exposing proteins.

In description you wrote about four systems but described are in this part only three (macroautophagy, microautophagy, and chaperone-assisted autophagy).

6. Your article is a review article and should give an overview of informations (data) from original papers with experimental data. From this point is not very correct that in some cases you used as an citation review articles.

Lines 391-392: “In general, a decline in proteasome function during senescence and aging has been observed in several mammalian tissues and cells [101]” Cited is Review Article: Chondrogianni, N.; Sakellari, M.; Lefaki, M.; Papaevgeniou, N.; Gonos, E.S. Proteasome activation delays aging in vitro and in vivo. Free Rad Biol Med, 2014, 71, 303-320.

Author Response

We are coming back with revised manuscript titled “Molecular chaperones and proteolytolitic machinery regulate protein homeostasis in aging cells” for further evaluation in Cells. We are thankful to reviewers for deep analysis of the manuscript. We carefully studied their comments and following the recommendations of Reviewers 2 and 3 we reformatted Table 1, corrected mistakes in the text and added a new part of Conclusions.  We believe that quality of the paper has benefited in result of the response to the comments.

Reviewer 2:

We thank the reviewer for the comments that we answering one by one:

Q1: You wrote (lines 76-77): “In this article, we briefly overview the function of proteostasis mechanisms in cell aging with an emphasis on molecular chaperones and their interrelationships with the other two systems.”

 The expression “with other two systems” should be described here exactly.  

R1: Thank you, we have corrected the text

Q2: You wrote (lines 92-93): “On the other hand, the stress response mechanism is essential for the normal development of a variety of organisms from C. elegans to mice.”

Do you mean that stress response mechanism is essential for the normal development only to mice? Is it not the truth for human?

R2: Yes, of course, it is true for humans. We have corrected.

Q3: You wrote (lines 79-80):“Chaperones, most of which belong to heat shock protein families Hsp100, Hsp90, Hsp70, Hsp60 79 and Hsp27 (small Hsps [sHsps]), are expressed …”

Hsp27 is a member of family of small Hsps. From this point is more correct to write families of  Hsp100, Hsp90, Hsp70, Hsp60 79 and small Hsps (such as Hsp27).

R3: Thank you, we have corrected.

Q4. Table 1 – the informations presented in this Table are not always clear. Table should sumarize information (data) about changes in expression of chaperones in aging and stressed cells. However, in my opininon is the structure of presented Table 1 not ideal and presented data are not complete. Some comments to this table: 

Q4a) It is not defined what exactly means in your Table + and ++.

R4a: We have added the notification what “+” and “++“mean in Table 1 in low row.

Q4b) As an example for my comment I choosed Hsp70 and data presented for rat brain, lung, skin (comment could be applied also for another Hsps and cells): For situation Y (young) is in response to stress for mRNA ++, for O (old) is for mRNA +. This indicates decline in mRNA levels. Were these changes in mRNA associated with protein levels modification? Are not available data about the Hsp70 expression at normal conditions?

R4b: We presented the results existing in literature so far including those obtained in 90th, when there was no antibodies recognizing all Hsps of different species. Unfortunately, today there is still no works unifying the data on Hsps mRNA and protein composition. So, our study appears to be the first attempt to gather these data. Clearly, it seems that a lot of work remains to complete the table.

Q5. Legend to the Figure 1.

There is sentence (lines 137-139): “Autophagy features at least four distinct protein degradation systems including macroautophagy that serves for digestion of polypeptides and organelles, microautophagy necessary for the degradation of useless membranous structures, and chaperone-assisted autophagy which targets KFERQ motif-exposing proteins.

In description you wrote about four systems but described are in this part only three (macroautophagy, microautophagy, and chaperone-assisted autophagy).

R5: Thank you, we have corrected the number of autophagy types

Q6:  Your article is a review article and should give an overview of informations (data) from original papers with experimental data. From this point is not very correct that in some cases you used as an citation review articles.

Lines 391-392: “In general, a decline in proteasome function during senescence and aging has been observed in several mammalian tissues and cells [101]” Cited is Review Article: Chondrogianni, N.; Sakellari, M.; Lefaki, M.; Papaevgeniou, N.; Gonos, E.S. Proteasome activation delays aging in vitro and in vivo. Free Rad Biol Med, 2014, 71, 303-320

R6: We agree with the reviewer’s comment and made appropriate changes in the text and added the relevant references.

Reviewer 3 Report

This is a timely and thoughtful review article on aging and innate cellular processes contributing to it and fighting it. Overall, the concepts covered and the details offered are important and will be of much use to the field. The issues are organizational and English. I have attached detailed comments in the file below. But, overall, chapters 4 and 5 seemed much smoother in organization and language, but the prior ones were a very tough read at times, since some of the concepts perhaps were improperly conveyed through word choices. There were also a ew rough parts where the article got derailed by philosophical considerations, which, while welcome and appreciated for their thoughtfulness, detracted and perhaps might fit better in a special chapter or section. I think that with some hard work at reorganizing it, including for example a conclusions chapter that can reframe the entire story and provide clues moving forward, this would be a welcomed review in the field. Details and comments in the PDF below. 

Author Response

We are coming back with revised manuscript titled “Molecular chaperones and proteolytolitic machinery regulate protein homeostasis in aging cells” for further evaluation in Cells. We are thankful to reviewers for deep analysis of the manuscript. We carefully studied their comments and following the recommendations of Reviewers 2 and 3 we reformatted Table 1, corrected mistakes in the text and added a new part of Conclusions.  We believe that quality of the paper has benefited in result of the response to the comments.

 Reviewer 3.

This is a timely and thoughtful review article on aging and innate cellular processes contributing to it and fighting it. Overall, the concepts covered and the details offered are important and will be of much use to the field. The issues are organizational and English. I have attached detailed comments in the file below. But, overall, chapters 4 and 5 seemed much smoother in organization and language, but the prior ones were a very tough read at times, since some of the concepts perhaps were improperly conveyed through word choices. There were also a ew rough parts where the article got derailed by philosophical considerations, which, while welcome and appreciated for their thoughtfulness, detracted and perhaps might fit better in a special chapter or section. I think that with some hard work at reorganizing it, including for example a conclusions chapter that can reframe the entire story and provide clues moving forward, this would be a welcomed review in the field. Details and comments in the PDF below

 We are grateful to the Reviewer 3 for a thorough analysis of our work. We were very attentive to his/her comments and made substantial amendments to the final text. We placed the answers to the comments that Reviewer made in the windows of the PDF file at the same place, in the "question-answer" mode.

Round 2

Reviewer 2 Report

Authors have done several modifications and corrections in the revised manuscript and included corresponding changes based on my comments.

Author Response

We thank the Reviewer 2 for favorable assessment of our correction of the manuscript.

Reviewer 3 Report

The authors have done a nice job with addressing my concerns. Overall, the review is in very good shape. There are grammatical and spelling errors that need careful attention, including several places where comma placements make the sentence hard to read, or give it the wrong "bend", the requiring further re-reads to know what is the meaning. But, outside of that, this will be a nice contribution to the field. The attached Pdf raises some language and clarification items that can help get it as close to a publication ready manuscript as possible. 

Author Response

We thank reviewer 3 for the positive estimating of our manuscript and we hope that the last proof reading allowed us to correct the remaining English roughness in the text.